# Exploration of Carbon Dioxide Curing of Low Reactive Alkali-Activated Fly Ash

**DOI:** 10.3390/ma15093357

**Published:** 2022-05-07

**Authors:** Peyman Harirchi, Mijia Yang

**Affiliations:** Department of Civil, Construction and Environmental Engineering, North Dakota State University, Fargo, ND 58104, USA; peyman.harirchi@ndsu.edu

**Keywords:** carbon dioxide curing, alkali-activated material, fly ash, FTIR, efflorescence, XRD

## Abstract

In this paper, the effect of carbon curing procedure on low reactive fly ash alkali-activated pastes was investigated. Specimens were cured with pure carbon dioxide (CO_2_) gas for different curing times under 4 bar pressure. Chemical and physical characteristics of the geopolymer pastes were obtained from mass monitoring, titration test, XRD, FTIR and TGA-DTG analyses. Regarding the test results, after three days of CO_2_ curing, the highest *CO*_2_
*uptake* was obtained at 4.8 wt% of fly ash precursor, with carbon sequestration efficiency at 22.6%. The ratio of carbon dioxide absorbed as efflorescence to the total absorbed CO_2_ was measured. The results show that at early age, almost 50% of carbonated products appeared as efflorescence; however, by increasing the curing time, and after 3 days of curing, about 80% of carbon dioxide was stored in the matrix. It was found that, in all cases, carbonation curing was detrimental to the geopolymerization process due to a high amount of efflorescence and led to a reduction in the compressive strength. At 24 h and 3 days, the specimens showed a lower reduction in compressive strength in comparison to CO_2_ samples cured at 3 h, 6 h and 12 h. Regarding the XRD results, calcite was detected in the 24 h and 3 days specimens, which contributes to lower pore sizes due to a higher molar volume and production of silica gel that might participate in the polymerization processes and results in densified microstructures.

## 1. Introduction

Due to calcination reaction of the carbonate minerals in cement manufacturing, the production of one tonne of cement approximately releases one tonne of carbon dioxide (CO_2_) [1]. Consequently, the cement industry has been responsible for almost 8% of CO_2_ emissions in 2020; therefore, the reduction and control of CO_2_ emission is becoming a major trend in the cement and concrete industry [2].

In recent years, CO_2_ curing of fresh concrete as a carbon capture and storage (CCS) method has been adopted due to the urgency of CO_2_ mitigation [3,4,5]. Using this technology, gaseous CO_2_ is permanently fixed into thermodynamically stable minerals, such as calcium or magnesium carbonates [6,7], which leads to improvement of physical properties and durability of concrete [3,8]. The effect of carbon curing generally depends on the rate of CO_2_ diffusion, carbonation rate, concentration of carbon dioxide, relative humidity and chemical composition of the mix [9,10,11,12]. The enhancement of the properties of final products is rooted in the quality and quantity of calcium carbonate generated in the pore structure in the early ages of hydration.

Pro-environment alternatives to the ordinary Portland cement (OPC) have been developed since 1940 called alkali-activated materials (AAMs) and geopolymer concrete (GC) [13]. AAMs are aluminosilicate industry byproducts, such as granulated blast furnace slag (GBFS) or fly ash (FA), which show cementitious properties by adding alkali solutions, such as sodium hydroxide (NaOH) and water glass [8]. The reaction forms a three-dimensional Si-O-(Si or Al) framework, and the final product contains sodium-aluminosilicate-hydrate (N-A-S-H) and calcium-aluminosilicate-hydrate (C-A-S-H) gels. The resulting solid has a rapid strength gain, low permeability, high chemical, high heat resistance and low thermal conductivity, which make these materials qualified candidates for replacing OPC with a significant reduction in CO_2_ [8,14,15,16,17].

The conventional curing methods of FA-based concrete are water curing and steam curing in which curing time and temperature are the most important factors for controlling the compressive strength [18]. Recent studies have investigated the effect of carbon dioxide curing on both OPC and AAMs for further reduction in CO_2_ footprint. Zhang et al. investigated the effect of CO_2_ curing on the pozzolanic reaction in fly ash concrete [7]. They found that if carbonation is limited to 12 h, it activates the fly ash OPC specimen at an early time of hydration, and the pH value of the pore solution is comparable to the control sample without the fly ash. Kassim et al. measured the effect of carbonation curing on the mechanical properties of alkali-activated electric arc furnace slag and reported a significant improvement of physical properties in CO_2_ curded samples [19]. Park et al. examined the effect of CO_2_ rich environment on FA-based alkali-activated materials with oven heating treatment [20]. They observed that the carbonated minerals densify the microstructure and improve the compressive strength of CO_2_ cured samples. It was found that in CO_2_ cured samples, the aluminosilicate gel contained a higher amount of silicates. Mei et al. investigated the mechanical and microstructure properties of alkaline-activated blast furnace slag (BFS) under accelerated carbonation [2]. They concluded that the carbonation of AAMs under a high CO_2_ concentration deteriorates the matrix structure due to the consumption of Ca ions in C-S-H gel. Consequently, the resulting pores in carbonated specimens accelerate CO_2_ diffusion and contribute to the weakening of the structure. Ohenoja et al. used peat-wood fly ashes from different sources for mineral carbonation [6]. The results showed an inconsistent effect of carbonation curing on the mechanical properties and a reduction in compressive strength in one fly ash sample, while in other samples, an increase in compressive strength was observed. Yamazaki et al. reported that in fly ash AAM, the N-A-S-H structure does not change with respect to accelerated carbonation [8]. Ul Haq et al. enhanced the fly-ash-based GC by in situ carbonation and reported that higher geopolymerization was achieved in the carbonated specimens [15]. Similarly, the polymerization effect of carbonation was reported by Nedeljkovic et al. [12].

The effect of CO_2_ on the geopolymerization process and the effectiveness of carbon mineralization require further investigation due to the inconsistency of reported results in the literature. Annually, large-volume fly ash fails to meet the requirements of national standards, such as ASTM C618, due to low reactivity [21]. In this paper, the effect of carbonation curing on alkali-activated pastes composed of low reactive fly ash without heat treatment is examined. The CO_2_ mineralization capacity is measured through titration tests for specimens cured in a carbonation chamber under 4 bar pressure of pure CO_2_. The mechanical properties of these samples are then tested, and the effect of carbon curing is investigated by XRD, FTIR and TGA analyses.

## 2. Materials

The chemical composition of the fly ash used in this study is measured by a Rigaku Supermini 200 X-ray fluorescence spectrometer (Applied Rigaku Technologies, Inc., Austin, TX, USA) and shown in Table 1. The composition indicates that the fly ash contains high amounts of SiO_2_, Al_2_O_3_ and CaO, and is categorized under type F regarding the ASTMC618-19 standard [22]. The basicity coefficient ($K_{b}=\frac{\mathrm{CaO}+\mathrm{MgO}}{{\mathrm{SiO}}_{2}+{\mathrm{Al}}_{2}O_{3}}$) and the hydration coefficient ($H_{M}=\frac{\mathrm{CaO}+\mathrm{MgO}+{\mathrm{Al}}_{2}O_{3}}{{\mathrm{SiO}}_{2}}$) are calculated as 0.27 and 0.78, respectively. The majority minerals found by the X-ray diffraction test are quartz, mullite and hematite, as shown in Figure 1. The surface area calculated through the Brunauer–Emmett–Teller (BET) method is 0.5034 m^2^/g.

The alkali activator used in this study is a mixture of sodium hydroxide (NaOH) with a purity of 98(%) and sodium silicate (Na_2_SiO_3_) solution with 9% Na_2_O and 28% SiO_2_. An amount of 5 M/L solution of NaOH is mixed with Na_2_SiO_3_ solution in a way that the molar ratio of silicon oxide to sodium oxide equals 1 (SiO_2_/Na_2_O = 1). The alkali activator is prepared 24 h before being added to the fly ash precursor and kept at laboratory temperature.

### 2.1. Preparation of Samples and Experimental Set-Up

The water to binder ratio (*w*/*b*) is set to 0.3 for all samples. Besides the added water in the activator, water of the sodium silicate solution is considered in the calculation of water content. For each curing time, three samples are prepared. The compressive strength test is conducted with respect to ASTM C305-20 using 2-inch cubes [23]. After demolding, the samples are cured in the carbonation chamber under 4 bar pressure of 100% CO_2_ in a set-up shown in Figure 2. Non-carbonation samples (NC) are cured in an ambient environment with 66% relative humidity (RH) at 22 °C temperature. The samples are cured for 3, 6, 12, 24 h and 3 days, and are named 3 h, 6 h, 12 h, 24 h and 3 days, respectively.

### 2.2. Characterization Methods

#### 2.2.1. Mass Monitoring

For measuring CO_2_ absorption, the mass monitoring method as a nondestructive method was adopted by previous researchers [24,25,26,27]. The key in measuring the *CO*_2_
*uptake* in carbonated samples is the estimation of water evaporation; therefore, a comparison between NC samples and carbonated samples is made for measuring the evaporable water. First, the water loss rate for NC samples (*WRNC*) is measured using Equation (1), and then, the change of mass with respect to CO_2_ curing is calculated by assuming the same evaporable water content for the carbonated samples. Based on ASTM C566-19, the samples are heated in an oven at a temperature 110 ± 5 °C to measure the mass of evaporable water [28].
(1)$$\begin{array}{c} WRNC=\frac{M_{1}-M_{2}}{M_{1}}\times 100 \end{array}$$

In Equation (1), *M*_1_ and *M*_2_ are mass of a sample before and after oven heating, respectively. The mass of evaporable water (*M_w_*) in all samples is calculated using *WRNC* and the initial mass of a sample (*M*_1_) by applying Equation (2).
(2)$$\begin{array}{c} M_{w}=\frac{M_{1}\times WRNC}{100} \end{array}$$

The *CO*_2_
*uptake* for carbonated samples is calculated using Equation (3) in which *M_b_* is the mass of fly ash used in each sample.
(3)$$\begin{array}{c} CO_{2}\;uptake\left(\%\right)=\frac{\left(M_{2}+M_{W}\right)-M_{1}}{M_{b}}\times 100\% \end{array}$$

#### 2.2.2. Measurement of Absorbed CO_2_

One important issue in the implementation of geopolymer concrete is efflorescence on the surface of specimens. Unreacted alkali ions diffuse on the surface, react with CO_2_ in the air and form white carbonates known as efflorescence [26]. The reduction in efflorescence can be investigated by the surface modification method [29], through the control of concrete microstructure and decrease in the fluidity of alkali [30]. Efflorescence can be evaluated in different ways, such as immersing the specimens in water and measuring the weight of dissolved salts after drying the solution [31]. In another method, the efflorescence image is captured using a camera and compared with Na^+^ concentration at a different pH value of leachate [30]. However, the image-based method is not accurate because the efflorescence products include calcium and potassium cations in addition to sodium cations [26,32].

Additionally, the measurement of OH^−^ and HCO_3_^−^ must be considered in the evaluation of efflorescence. Due to lack of efflorescence measurement standards in AAMs, the water immersion method is adopted in this study, since carbonation products, including sodium carbonates and bicarbonates, are soluble in water. After CO_2_ curing, the samples are immersed in water for 2 days. In this process, water dissolves sodium carbonates and bicarbonates on the surface, but the calcium carbonate fixed in the matrix remains insoluble. The concentration of carbonate minerals in the solution is obtained by applying the titration test following the requirements of ASTM D3875-15 [33]. The ratio of CO_2_ in the soluble products to the total CO_2_ absorption shows the efficiency of carbonation curing.

#### 2.2.3. Compressive Strength

The samples, with respect to different carbon curing durations, are tested by the universal compression testing machine. The low reactivity of fly ash indicates that only late-age compressive strength is detectable; therefore, a 28-day compressive strength is reported for all samples regarding ASTM C109/C109M [34]. For each CO_2_ curing configuration, three samples are prepared, and the average compressive strength is plotted as a bar chart with the error bar representing the range of the strength variation.

#### 2.2.4. X-ray Diffraction (XRD)

XRD diffractograms are conducted using Rigaku SmartLab (Applied Rigaku Technologies, Inc., Austin, TX, USA) for identification of minerals and phase changes of crystals in all samples [8]. XRD test shows the crystalline phases for samples with respect to different carbonation times. Different polymorphs of calcium carbonates could be detected at high pressure of CO_2_. The peaks of calcium carbonates, including calcite (higher stability and crystallinity) and vaterite (lower stability and crystallinity), are expected in this experiment.

#### 2.2.5. Fourier Transform Infrared Spectroscopy (FTIR)

FTIR measures the absorption of infrared radiation by different functional groups; it identifies the molecular components and structures in the material [35]. FTIR is applied to specimens through the Thermo Scientific Nicolet 8700 FT-IR spectrometer (BRUKER, Allentown, PA) to monitor the change of the gel structure for different carbon-cured samples. FTIR is also conducted for measuring the reactivity of fly ash by the method proposed by Zhang et al. [36]. By deconvolution of the absorbance spectrum between 400 cm^−1^ and 1400 cm^−1^ band, the Gaussian curves corresponding to the active and inactive bonds are obtained. In the active bonds, Si and Al atoms are easily dissolved in the geopolymerization process. The active bonds are represented in Table 2. The relative area of convoluted bonds is assumed as an index of concentration of each bond. Since the solubility of active bonds is not equal, reactivity coefficients are applied to modify the contribution of each active bond. Regarding Ref [36], weaker bonds appear in the lower regions of the FTIR spectrum; consequently, reactivity coefficients for high to low regions are chosen as 0.25, 0.5, 0.75 and 2.

#### 2.2.6. Thermogravimetric Analysis (TGA)

TGA is performed on all samples using TA Instrument Q500 to analyze their chemical compositions. The samples are heated with pure nitrogen gas from 25 °C to 1000 °C at the rate of 20 °C/min to quantify both carbonated and amorphous minerals.

## 3. Results and Discussions

The test results of the CO_2_ cured samples are presented in this section. First, the reactivity of the fly ash samples is measured using FTIR, and then, the mechanical properties, including mass change, compressive strength and amount of CO_2_ absorption, are presented. Next, spectroscopic results of the CO_2_ cured samples and NC samples are demonstrated by XRD, FTIR and TGA. 

### 3.1. Quantitative Measurement of Reactivity of Fly Ash

Figure 3 shows the FTIR absorbance and Gaussian convoluted curves of the active bonds. It is assumed that the relative area of a resolved bond represents its concentration in the sample. The reactive surface area is calculated by multiplication of the active concentration by surface area obtained from the BET gas adsorption test. The relative area, reactive coefficient, active concentration and reactive surface area of the FA sample are shown in Table 3.

The active bond represents the Si-O-Al bond, which is easily broken in the presence of an alkali activator and dissolves at a faster rate in comparison to the inactive bonds. These bonds are mostly related to non-bridging oxygen with the Q^3^, Q^2^, Q^1^ and Q^0^ molecular structure [36]. The reactive surface value (0.144 m^2^/g), in comparison to the values reported in Ref [36], indicates low reactivity of the FA sample in this study.

### 3.2. Mass Change of Carbon-Cured Geopolymer Paste and the CO_2_ Absorption Capacity

The mass change after oven heating shows the amount of evaporable water for NC samples. The evaporable water is measured as 18.1% of the initial mass of the specimen. The mass change for CO_2_ cured sample is lower compared to NC samples because of CO_2_ absorption (Figure 4). The reduction in mass change by increasing the curing time indicates more absorption of CO_2_.

*CO*_2_*uptake* per mass of binder is calculated using Equation (3) and the average evaporable water in NC samples. The *CO*_2_
*uptake* at the early age occurs rapidly, and the rate of *CO*_2_
*uptake* for the first 3 h is about 0.46%/h. However, the *CO*_2_
*uptake* rate decreases between 3 and 24 h (Figure 4). This might be because of the presence of carbonated products that precipitate on the surface and in the pore structure, which hinders the diffusion of gaseous CO_2_ into the samples. By further CO_2_ curing, the gas molecules diffuse deeply into the matrix, and the precipitation rate of carbonated products increases.

The maximum capacity of CO_2_ absorption can be determined by the chemical composition of cementitious materials through Equation (4) [37,38]:(4)$$\begin{array}{c} {\mathrm{CO}}_{2}=0.785\left(\mathrm{CaO}-0.7{\mathrm{SO}}_{3}\right)+1.091\mathrm{MgO}+1.42{\mathrm{Na}}_{2}O+0.935K_{2}O \end{array}$$

It should be noted that the above formula was suggested for OPC. Consequently, caution should be taken when using Equation (4) for alkali-activated materials [37]. Based on Equation (4), the maximum CO_2_ absorption capacity of the specimens is 21.23%. The maximum *CO*_2_
*uptake* after 3-day CO_2_ curing is approximately 5% from the test data, which indicates a major portion of the samples is not carbonated. Depending on the source of the fly ash, different values of *CO*_2_
*uptake* efficiency have been reported. Hernandez et al. proposed the aqueous mineralization of carbon dioxide in fly ash powder without addition of an activator and calculated a 2.6% *CO*_2_
*uptake* per tonne of fly ash [39]. Calcium hydroxide carbonation was considered the main reaction controlling the mineral sequestration of CO_2_. The same reaction mechanism was considered by Mazzella et al. who obtained *CO*_2_
*uptake* at about 18 wt% fly ash by gas–solid carbonation treatment [40].

### 3.3. Efflorescence Measurement

CO_2_ cured samples show a high amount of efflorescence. This phenomenon deteriorates both the appearance aesthetics and mechanical properties of AAMs. High amount of efflorescence is observed on the surfaces, especially on the top of the samples.

Regarding the titration test, only carbonate and hydroxide ions are observed, while bicarbonate ions are not detected. By measuring the concentration of carbonate ions, the weight of hydrated carbon dioxide in the solution is calculated. Figure 5 represents the *CO*_2_
*uptake* and its rate per mass of the paste versus curing time obtained from the titration test. The absorption rate of carbon dioxide indicates that in the early stages of curing, the paste has a high absorption capacity, while this potential decreases significantly over time.

The efficiency of CO_2_ mineralization mentioned in the literature for aqueous carbonation and gas–solid carbonation is 82% and 74%, respectively [39,40]. These efficiencies are obtained for powder samples, while the CO_2_ mineralization examined here is for the paste specimens. The efficiency ratios for 3 h, 6 h, 12 h, 24 h and 3 days specimens are obtained by dividing the *CO*_2_
*uptakes* from Figure 5 by the total capacity of carbonation calculated from Equation (4), which are 6.6, 9.4, 11.3, 12.7 and 22.6%, respectively. The ratio of CO_2_ absorbed as efflorescence to the total absorbed CO_2_ shows the effectiveness of carbon curing (Figure 6). In the early age of carbonation, almost half of the carbonated products appear as efflorescence due to the existence of a high amount of free alkaline ions on the sample surface, while at longer curing times, more CO_2_ is fixed in the matrix. The dissolution of CO_2_ forms carbonic acid in the pore solution and may cause a reduction in the pH of the pore solution, which occurs in natural carbonation and hinders further progress of the reaction. In OPC, calcium hydroxide acts as a buffer and saturates the pore solution with Ca^2+^ and OH^−^; consequently, the pH level of the pore solution stays above 12, which protects the steel reinforcement against corrosion. CO_2_ buffer capacity significantly affects the carbonation resistance and can be represented as the ratio of water to reactive calcium oxide [10]. In severe cases, after the consumption of portlandite, the pH of the pore solution starts to decrease. By lowering the pH under 12.6, C-S-H, ettringite (AFt) and monosulfate (AFm) become unstable. Additionally, the protection layer of steel reinforcement disappears when the pH is less than 9.5, and eventually, calcium carbonates precipitate in the pore structure. In AAMs, carbonation occurs in a two-step process:(1)Precipitation of sodium carbonates and reduction in pH of the pore solution.(2)Consumption of calcium-rich products.

The carbonation of AAMs, unlike OPC, depends on both the reactive CaO and Na_2_O content of the precursor [12]. In comparison to OPC, AAMs do not include portlandite and contain low amount of reactive calcium. The lack of portlandite results in the decalcification of C-A-S-H gel and the production of silica gel that might be engaged in the geopolymerization process and densify the AAMs’ microstructure.

### 3.4. Compression Strength Results

The change of the 28 days’ strength is monitored for NC and CO_2_ cured samples in Figure 7. The CO_2_ curing is found to be detrimental in all cases in comparison to NC samples. However, in 24 h and 3 days specimens, higher compressive strength was obtained compared to 3 h, 6 h and 12 h samples. This is probably due to the existence of calcium carbonates in the pores and the participation of the produced silicates in the polymerization process. After a 12 h curing, CO_2_ is absorbed as a stable carbonate product, such as calcium carbonate, with different morphologies, including calcite, aragonite and vaterite. Calcium silicates are mostly converted to calcium carbonates based on the following reaction in carbon-cured samples [15]:(5)$$\begin{array}{c} \;{\mathrm{CaSiO}}_{3}+{\mathrm{CO}}_{2}⟶{\mathrm{CaCO}}_{3}+{\mathrm{SiO}}_{2}\; \end{array}$$

The produced silicate reacts with the silicate network in the geopolymer and improves the structure of the matrix. The polymerization of the aluminosilicate gel and increasing molar volume of products, which reduces the porosity, have been reported in previous literature [41,42]. Unlike OPC, where the production of calcium carbonate (CaCO_3_) is detrimental during carbonation, in GC, this phenomenon could have the opposite effect because the main structure in AAMs is constructed based on silicates and aluminates [15], and the additional SiO_2_ produced through the reaction explained by Equation (5) might supply more reactive silica and enhance the geopolymerization process. This might be the reason for the increased strength in 24 h and 3 days samples. The presence of calcite in 24 h and 3 days specimens detected by XRD is further evidence that could explain this discrepancy in compressive strength results.

### 3.5. XRD Test Results

XRD analysis is conducted to analyze the products of CO_2_ curing geopolymer with respect to the different curing durations (Figure 8). According to the XRD patterns, 24 h and 3 days samples are similar with respect to the identified minerals. The calcite peaks (2θ = 29.6°, 39.5°, 43.6°, 47.2°) [8] are detected in both samples. The presence of calcite, which precipitates in the pores, might be the reason for the increased strength after 24 h CO_2_ curing. The formation of calcite and vaterite after one day in alkaline-activated blast furnace slag (BFSS) has also been reported in the literature [2]. Vaterite is unstable under accelerated carbonation and converts to calcite; however, the growth of calcite depends on the adequate supply of CO_2_ and dissolution rate of Vaterite [2]. Gaylussite (Na_2_Ca(CO_3_)_2_·5H_2_O.) is detected in both 24 h and 3 days samples. Natron is a carbonation product of low CO_2_ concentration, and nahcolite is a carbonation product of high concentration of carbon dioxide [41]. In the obtained XRD patterns, nahcolite (2θ = 30.58°, 34.58°, 40.84°) peaks might be overlapped with gaylussite peaks. In the initial steps of carbonation, vaterite is expected to form as a metastable polymorph of calcium carbonate [43,44]. Quartz, mullite and hematite are identified in the samples as well [45]. In the presence of high humidity, vaterite crystallizes into calcite [46]. Besides the moisture content, the pH value of pore solution also contributes to the crystallization of vaterite. The pH value of pore solutions in AAMs after carbonation is usually above 9 in natural carbonation [47]. C(N)-A-S-H gel is thermodynamically less stable than C-S-H gel and is mainly amorphous, which causes different phase dissolution resulting from carbonation compared to OPC [48,49]. Pirssonite (Na_2_Ca(CO_3_)_2_ · 2H_2_O) and calcium disilicate (CaSi_2_O_5_) are detected in 3 h, 6 h and 12 h NC samples.

### 3.6. FTIR Test Results

The infrared spectra for NC and CO_2_ cured samples are shown in Figure 9. The absorption bands are divided into three regions, including water-related bonds, Si-O-Si bond and carbonate-related bonds [12]. In geopolymers, the amount of the non-bridging oxygen (NBO) depends on the SiO_2_/Na_2_O ratio of the activator solution [50]. When SiO_2_/Na_2_O ratio decreases, NBO decreases and contributes to lower silicon coordination with oxygen, such as the SiQ^2^ and SiQ^1^ units. Q stands for the oxygen bond formed with Si, and numbers 1 and 2 stand for the number of bonds formed. The geopolymerization process is recognized by Si-O-T (T: Si or Al) bonds with a wave number at 950 cm^−1^ to 1200 cm^−1^ and at 650 cm^−1^ to 750 cm^−1^. Bands at 450 cm^−1^ to 1300 cm^−1^ are related to the Si-O-Si bond, and bands between 1600 cm^−1^ and 4000 cm^−1^ represent chemically bonded water. The stretching vibrational bonds of Si-O-Si are infrared active. The silicon coordination number, including SiQ^4^, SiQ^3^, SiQ^2^, SiQ^1^ and SiQ^0^, is identified by wavenumbers at 1200, 1100, 950, 900 and 850 cm^−1^, respectively [35,43,50,51]. In all samples, it is seen that the absorption bands related to Q^3^ and Q^2^ are dominant. 

The 713 cm^−1^ absorption peak that appears in the 12 h, 24 h and 3 days samples shows the bending vibration of CO_3_^2−^ in calcite and aragonite [2,43]. The increase in band intensity at this wavenumber can be considered as the condensation of aluminosilicate gel with a high amount of aluminum. The absorbance that appears between 1400 cm^−1^ and 1500 cm^−1^ is related to the stretching of CO_3_^2−^ and represents the existence of calcite and vaterite [35,52,53]. It is seen that by increasing the CO_2_ curing time, the intensity of the band increases. At the early-age CO_2_ curing, the existence of aragonite and vaterite for NC, 3 h, 6 h samples is observed at 1489 cm^−1^ and 1490 cm^−1^. The existence of this band for non-carbonated samples shows the effect of carbonation weathering. The presence of nahcolite (NaHCO_3_) is seen at band 1450 cm^−1^, and calcite is detected at band 1419 cm^−1^ [12] for 12 h, 24 h and 3 days samples, which is consistent with the XRD results. As CO_2_ curing progresses, more crystalized CaCO_3_ forms. The out-of-plane bending mode of the carbonates is located at 875 cm^−1^ [35], which is only detected for 12 h, 24 h and 3 days samples.

In 12 h, 24 h and 3 days samples, a peak close to 2950 cm^−1^ is seen, which could possibly indicates the existence of vibrational bonds between carbon and hydrogen that has been detected in stearic acid [54]. The bending vibration of H-OH bonds appears in 1640 cm^−1^, which shows the existence of chemically bonded water [12,35]. It is seen that the peak is almost unchanged among all samples. The absorption at 3445 cm^−1^ shows the stretching vibration of the O-H bond in bonded water [2]. The absorption of bonded water in calcium hydroxide appears at 3644 cm^−1^ [2,55,56], which has low intensity.

### 3.7. TGA Test Results

The results of thermogravimetric analysis (TGA) and its derivative to temperature (DTG) are represented in Figure 10. In the NC sample, the predominant mass loss occurs below 400 °C. The change of mass between 25 °C and 105 °C shows the loss of evaporable water. The mass change from 105 °C to 215 °C shows the loss of bonded water in matrix and decomposition of C(N)-A-S-H gel [8]. The amount of C(N)-A-S-H gel decreases from NC samples to 3 days samples. The surface area under the DTG curves shows that CO_2_ cured samples have a lower amount of bonded water in their structure [35,57]. The peak at 170 °C is related to thermal decomposition of hydrotalcite [58]. The dehydration of pirssonite and gaylussite occurs at temperatures under 250 °C [59]. Natron is dehydrated at 160 °C [60] in CO_2_ cured samples, and the intensity of the peak reduces as curing progresses. The decomposition of total carbonated phases occurs in the range of 400 °C to 800 °C [35,42,61]. Amorphous calcium carbonate (CaCO_3_) decomposes at 400–600 °C [8]. Higher amount of crystalline CaCO_3_ is detected (600–750 °C [8]) in the case of increased CO_2_ curing time [62]. However, the morphology of calcium carbonates in FA-based AAMs is more amorphous [12]. Regarding the literature, the crystallinity of polymorphs of calcium carbonates depends on many factors, including temperature, relative humidity, chemical composition of the binder and concentration of carbon dioxide [12,63].

## 4. Conclusions

In this study, the effect of CO_2_ curing on low reactive fly ash alkali-activated pastes in the early ages was investigated. Five CO_2_ curing times were applied on alkaline-activated FA samples in the presence of pure CO_2_ under four-bar pressure. The following conclusions are drawn according to the experimental results:Curing time had a significant effect on the carbonation products and the amount of absorbed CO_2_. The maximum *CO*_2_
*uptake* was obtained for the 3 days sample at 4.8 wt% fly ash, and the maximum efficiency was obtained at 22.6%.In short-term curing times, almost 50% of the absorbed carbon dioxide was efflorescence; however, by increasing the curing time, the absorbed CO_2_ in the matrix increased in the form of insoluble and stable products.It was observed that carbon curing had a detrimental effect on the compressive strength in all specimens in comparison to the control sample. The presence of carbonic acid consumed the alkali and hydroxide ions in the pore solution, which lowered the pH of the pore solution and, consequently, hindered the progress of the geopolymerization reaction. At later ages of curing (after 24 h), the compressive strength started to rise slightly, which could be due to the presence of calcite and silica gel in the pore structure that was detected by XRD in 24 h and 3 days specimens.

## Figures and Tables

**Figure 1 materials-15-03357-f001:**
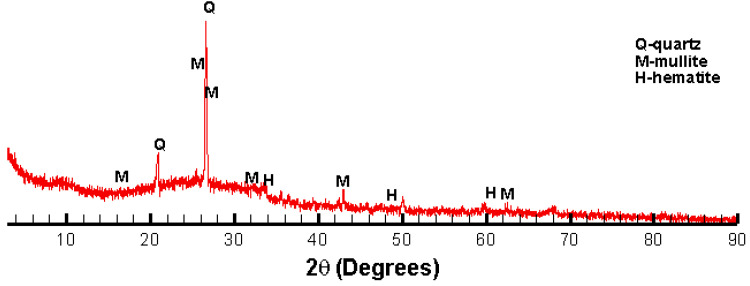
XRD results of the used fly ash sample.

**Figure 2 materials-15-03357-f002:**
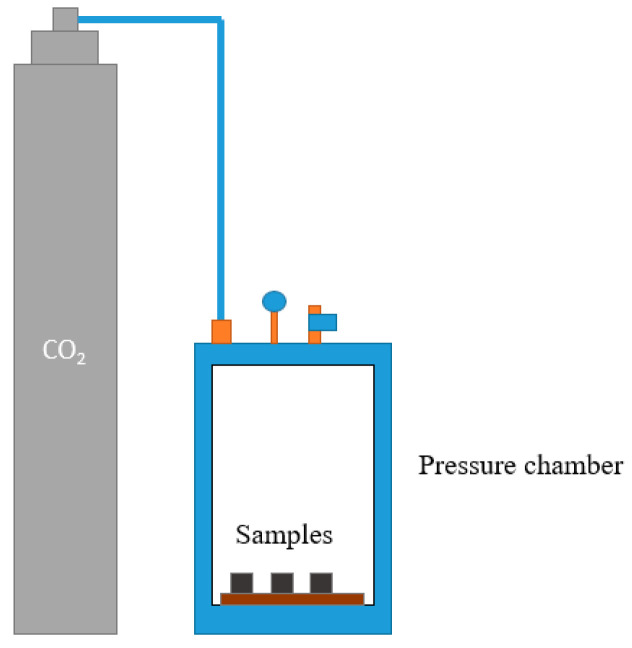
Carbon curing set-up.

**Figure 3 materials-15-03357-f003:**
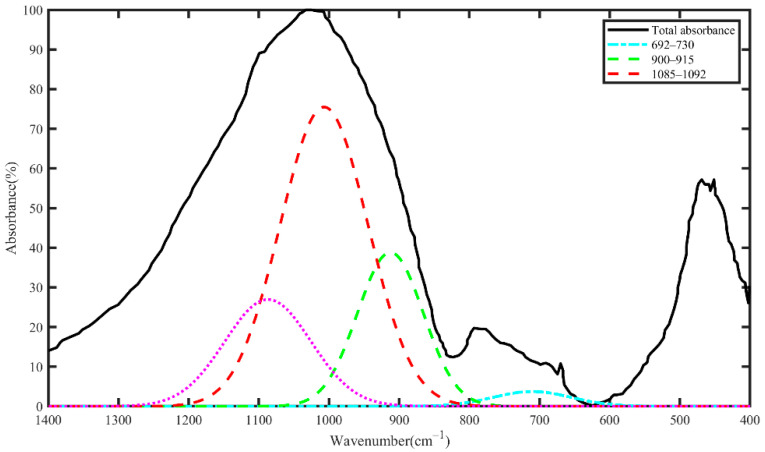
FTIR absorbance and Gaussian curves of active bonds in the FA sample.

**Figure 4 materials-15-03357-f004:**
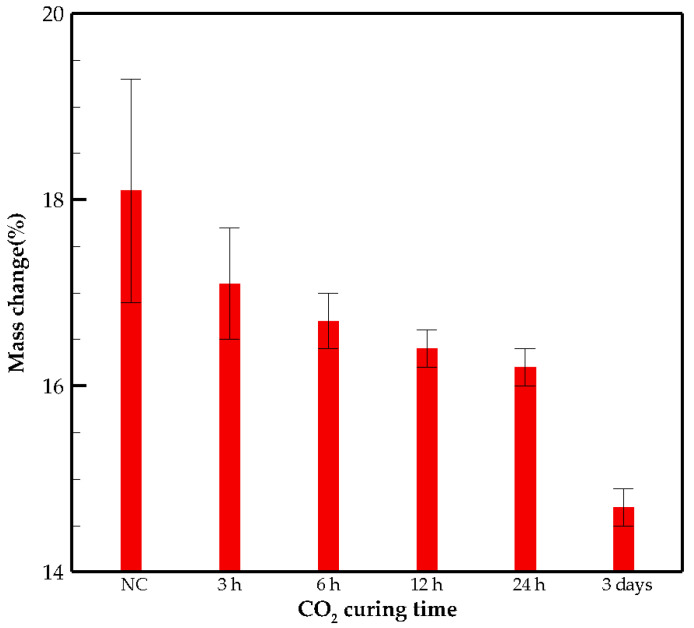
Mass change of samples after oven heating (%).

**Figure 5 materials-15-03357-f005:**
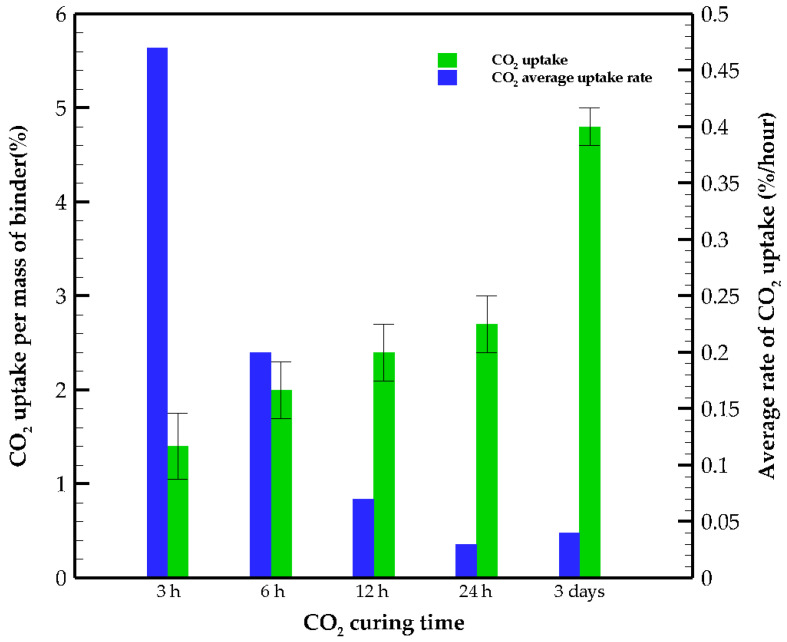
*CO*_2_*uptake* per mass of FA and its average rate with respect to curing time for specimens.

**Figure 6 materials-15-03357-f006:**
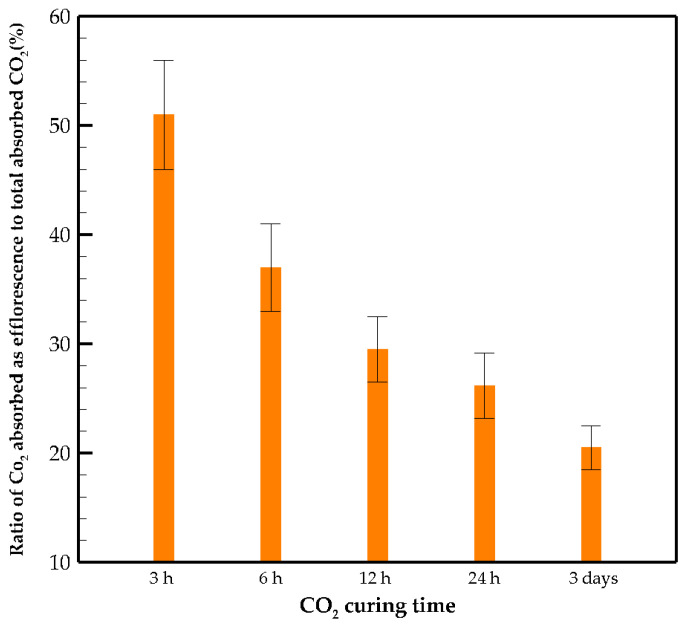
Percentage of CO_2_ absorbed as efflorescence with respect to curing time.

**Figure 7 materials-15-03357-f007:**
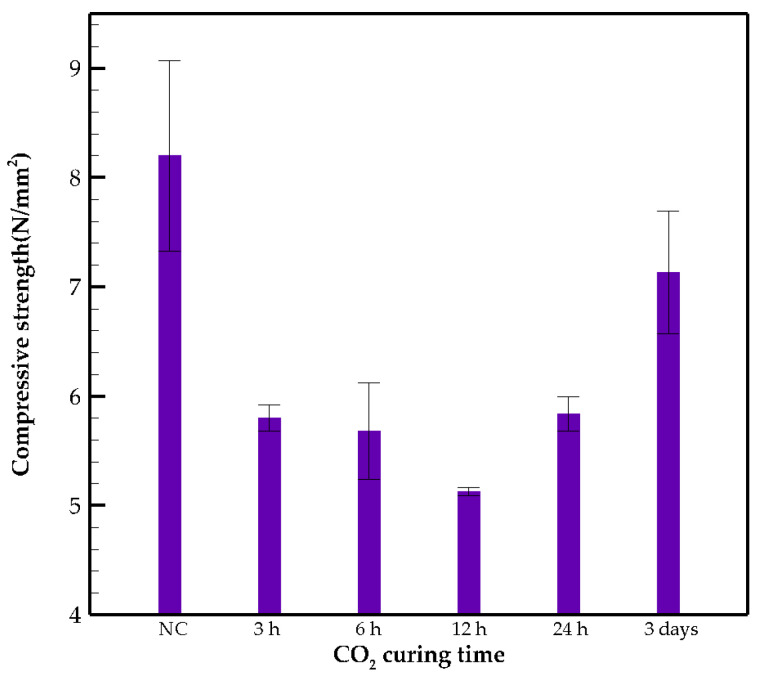
28 days’ compressive strength.

**Figure 8 materials-15-03357-f008:**
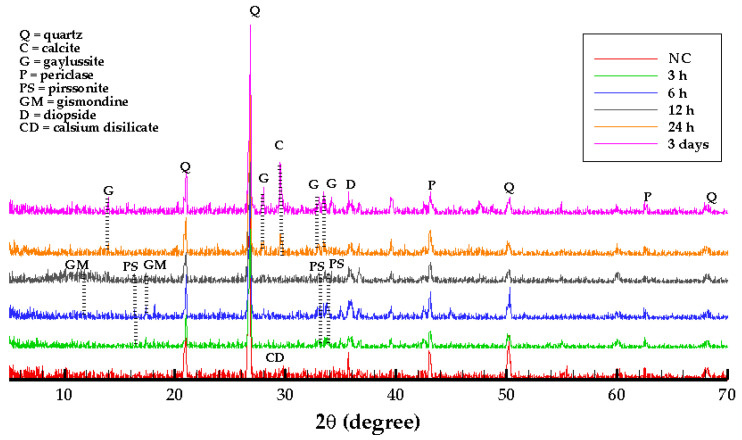
XRD diffractograms of non-carbonated and carbonated samples.

**Figure 9 materials-15-03357-f009:**
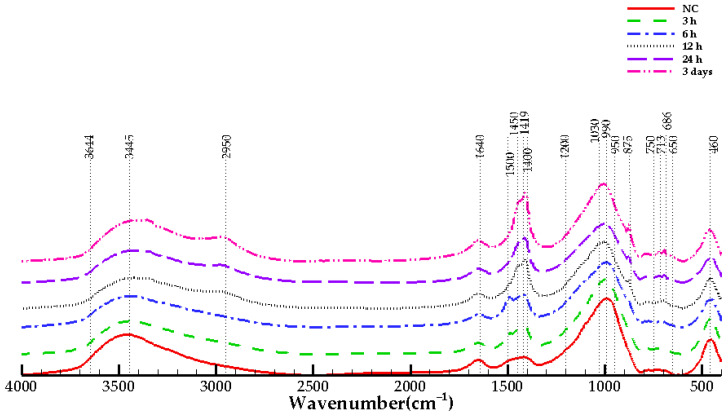
FTIR spectroscopy of NC and CO_2_ cured samples.

**Figure 10 materials-15-03357-f010:**
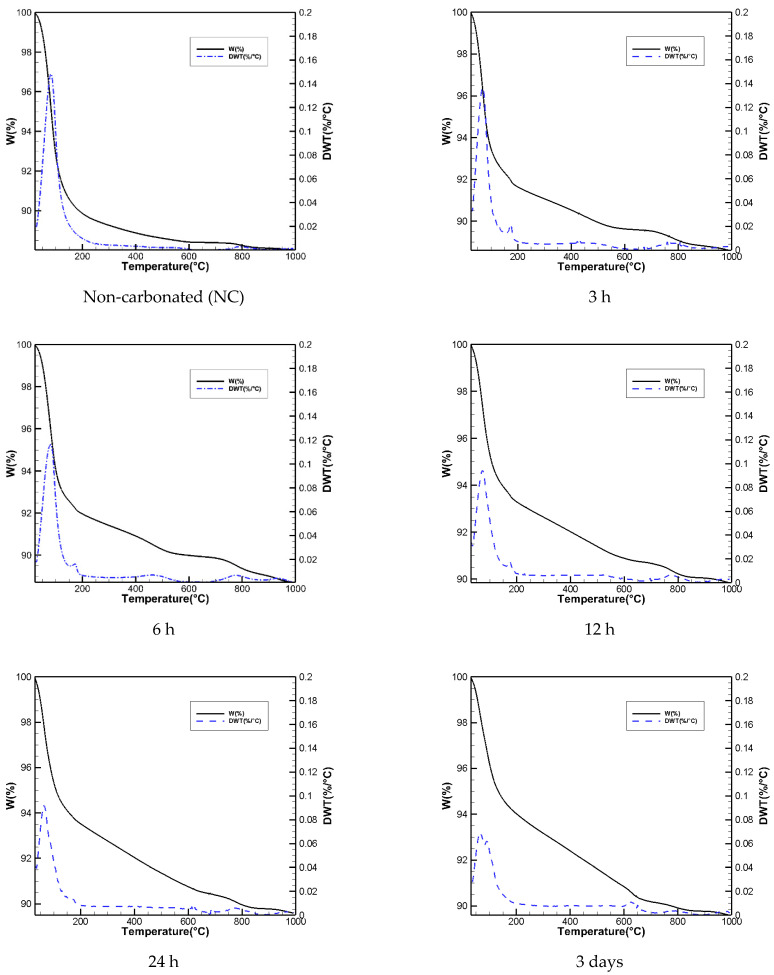
Thermogravimetric analysis results.

**Table 1 materials-15-03357-t001:** Chemical composition of the used fly ash.

	CaO	SiO_2_	Al_2_O_3_	MgO	Fe_2_O_3_	TiO_2_	K_2_O	Na_2_O	Other
Fly ash	13.7	51.1	16.2	4.2	6.1	0.7	2.64	2.85	2.51

**Table 2 materials-15-03357-t002:** FTIR range of active bonds in geopolymerization (adopted from Ref [36]).

Range	Bond Description
1085–1092	Asymmetric stretching of (Si, Al)-O-Si in glass phase, Q^3^
997–1011	Asymmetric stretching of (Si, Al)-O-Si in amorphous glass phase
900–915	Asymmetric stretching of Si-O^M^ where M is an alkali metal
692–730	Symmetric stretching of Al-O in (Si, Al)-O-Al

**Table 3 materials-15-03357-t003:** Reactive bond parameters and reactivity calculation of the FA sample.

Active Peak Center	Identified Bond	Relative Area (%)	Reactivity Coefficient	Active Concentration (%)
710.1	Al-O	1.37	2	2.74
912.5	Si-O-M	11.34	0.75	8.5
1007	(Si, Al)-O-Si	29.3	0.5	14.7
1087	(Si, Al)-O-Si	10.3	0.25	2.6
Total active concentration				28.54
Reactive surface area (m^2^/g)				0.144

## Data Availability

Data sharing not applicable.

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
