# Peer review of "Exploration of Carbon Dioxide Curing of Low Reactive Alkali-Activated Fly Ash"

_materials, 2022, doi:10.3390/ma15093357_

Round 1
Reviewer 1 Report
Dear Authors,
The manuscript “Exploration of carbon dioxide curing of low reactive alkali-activated fly ash” is very good. The paper was written in standard, grammatically correct English, small corrections are necessary. The title is clear. The content is in accord with title. The manuscript can adhere to the journal's standards after revision. The size of the article is appropriate to the contents. The authors must underline the major findings of their work and explained novelty of this study. The Abstract is OK. The Abstract section refers to the study findings, methodologies, discussion as well as conclusion. The key words permit found article in the current registers or indexes. Secondary raw material is too generally. In the introduction isn’t clearly described the state of the art of the investigated problem. The references from last years are necessary. It is study actual? The methods are well described and the equipment. The materials have been adequately described. For each device were presented more details. The paper was written in standard, grammatically correct English, small corrections are necessary. The figures have a relatively good quality.The tables contain necessary results. It is necessary comparison with their published papers or other similar studies. Please provide comparison with other studies.The Conclusion is OK. The paper is easy to understand by readers from other area. The literature is insufficiently critical, current, and internationally evaluated. Please citation references from last years.
The paper is in Materials journal’s topics. Please provide 2 papers from this journal (last year 2022)
Reviewer 2 Report
The manuscript entitled ‘Exploration of carbon dioxide curing of low reactive alkali-activated fly ash’ is in line with the Materials journal. The topic is up-to-date and connected with important research that could influence the implementation the idea of circular economy, especially the reduction of carbon footprint. The article based on original research. The abstract is informative enough. Overall, the article is well composed and requires only minor revisions, including:
- All article: Please use a reference format consistent with the template.
- Introduction (second paragraph): please compare with some data for geopolymers, including: https://doi.org/10.3390/ma14195741
- Introduction: ‘Mei et al. …” – lack of reference behind citation.
- Introduction: Please, stress the novelty aspects in the presented research (last paragraph).
- Materials: XRF, XRD, and other equipment; please specify the type of equipment.
- Chapter 2.2.3. Please add information on how many samples were investigated in each series.
- Chapter 2.2.4, 2.2.5 and 2.2.6 – please include type of equipment.
- Table 3: add information about identified bounding.
- Chapter 3.2.: A wider discussion with the literature is required.
- Chapter 3.3: Compare efficiency with other methods such as wet curing and application additives, for example, coffer grounds.
- Figure 7: requires more in-depth comments. What kind of differences are between the samples and what caused this differences?
- Conclusions: Please add more detailed conclusion based on obtained measurable results.
Reviewer 3 Report
1. Why is there no numerical result in the abstract of the technical article?
2. It is better to arrange keywords in alphabetical order.
3. The introduction is written very poorly. In particular, the literature review should be expanded to cover the various modern types of building materials. This will significantly strengthen the position of this highly rated journal dedicated to materials. For example:
4. At the end of the introduction, the purpose of the study and the tasks to achieve it should be clearly indicated. And in the output, give a numbered list of the corresponding solved tasks.
5. In Figure 7, the overall boundary of the diagram (4 MPa) can be taken much lower. Same for figures 2 and 6
6. Why is there such a high peak of crystalline quartz in Figure 1 and there is no calcium oxide at all, which, according to Table 1, is almost 14%? By the way, this is probably a mistake, because there should be more calcium oxide than aluminum oxide. Otherwise, it is not fly ash, but slag.
Round 2
Reviewer 3 Report
Most of my comments have been taken into account. However, due to the fact that the manuscript is devoted to the creation of green composites and the disposal of various production wastes, I ask you to consider the possibility of expanding the literature review on this topic. For example:
-Rahal, V.F., Trezza, M.A., Tironi, A., Castellano, C.C., Pavlikova, M.,
Pokorny, J., Irassar, E.F:, Jankovsky, O., Pavlik, Z. Complex
Characterization and Behavior of Waste Fired Brick Powder-Portland
Cement System, Materials 12 (2019) 1650.
- Volodchenko, A.A., Lesovik, V.S., Cherepanova, I.A., Volodchenko A N, Zagorodnjuk L H, Elistratkin M Y Peculiarities of non-autoclaved lime wall materials production using clays. IOP Conference Series Materials Science and Engineering 2018. 327(2):022021. DOI: 10.1088/1757-899X/327/2/022021
- Amran, M.; Fediuk, R.; Murali, G.; Avudaiappan, S.; Ozbakkaloglu, T.; Vatin, N.; Karelina, M.; Klyuev, S.; Gholampour, A. Fly Ash-Based Eco-Efficient Concretes: A Comprehensive Review of the Short-Term Properties. Materials 2021, 14, 4264. https://doi.org/10.3390/ma14154264
